# Experimental and Numerical Analysis of Steel-Polypropylene Hybrid Fibre Reinforced Concrete Deep Beams

**DOI:** 10.3390/polym15102340

**Published:** 2023-05-17

**Authors:** Sarah Khaleel Ibrahim, Noor Abbas Hadi, Majid Movahedi Rad

**Affiliations:** 1Department of Structural and Geotechnical Engineering, Széchenyi István University, H-9026 Győr, Hungary; sarah.khaleel.ibrahim@hallgato.sze.hu; 2Department of Civil Engineering, Ashur University College, Baghdad 10011, Iraq; noor_abbas1992@yahoo.com

**Keywords:** deep beam, hybrid polymer reinforced concrete, STEEL fibres, polypropylene fibres, concrete damage plasticity model

## Abstract

This work experimentally and numerically explored how varied steel-polypropylene fibre mixtures affected simply supported reinforced concrete deep beams. Due to their better mechanical qualities and durability, fibre-reinforced polymer composites are becoming more popular in construction, with hybrid polymer-reinforced concrete (HPRC) promising to increase the strength and ductility of reinforced concrete structures. The study evaluated how different combinations of steel fibres (SF) and polypropylene fibres (PPF) affected beam behaviour experimentally and numerically. The study’s focus on deep beams, research of fibre combinations and percentages, and integration of experimental and numerical analysis provide unique insights. The two experimental deep beams were the same size and were composed of hybrid polymer concrete or normal concrete without fibres. Fibres increased deep beam strength and ductility in experiments. The calibrated concrete damage plasticity model in ABAQUS was used to numerically calibrate HPRC deep beams with different fibre combinations at varied percentages. Based on six experimental concrete mixtures, calibrated numerical models of deep beams with different material combinations were investigated. The numerical analysis confirmed that fibres increased deep beam strength and ductility. HPRC deep beams with fibre performed better than those without fibres in numerical analysis. The study also determined the best fibre percentage to improve deep beam behaviour where a combination of 0.75% SF and 0.25% PPF was recommended to enhance load-bearing capacity and crack distribution, while a higher content of PPF was suggested for reducing deflection.

## 1. Introduction

The deep reinforced concrete beams are commonly used, which are structural components with large depths relative to their span. They can withstand bending moments, shear stresses, and deflection because of their thick sections. Shear, flexure, and compression all interact, making for a challenging design environment when working with deep reinforced concrete beams. Strut-and-tie models or finite element analysis are commonly used when constructing deep beams out of reinforced concrete, with consideration given to factors such as the location and size of the supports, the type and amount of reinforcing steel, and the shear and flexural strengths of the concrete. Several models for estimating the shear strength of deep beams made of reinforced concrete (RC) with and without web reinforcement were analysed by Shahnewaz et al. [1]. Each model’s precision was measured against data from 381 deep-beam tests. According to the research, the shear strength of RC deep beams was overestimated using the current code equations. As a result, a genetic algorithm was used to suggest revised shear equations by determining the most influential parameters and their correlations with shear strength in deep beams. Shear failure of reinforced concrete deep beams was analysed by Ashour [2]. Calculations were made to determine the shear strength of deep beams based on the position of the instantaneous centre of relative rotation of the blocks in motion. Good agreement was found between the comparisons and the experiments. The shear-span-to-depth ratio was found to have a greater impact on the shear capacity than the span-to-depth ratio in parametric research. The shear capacity can be increased by adding bottom reinforcement, but only to a certain point, i.e., where the shear-span-to-depth ratio governed the proportional efficacy of horizontal and vertical web reinforcement. Abadel et al. [3] investigated how well shear stirrups and carbon fibre-reinforced plastic (CFRP) strengthening added to the shear strength of deep beams. There were six samples in total, and they were divided into three groups of two for testing. The results demonstrated that the shear strength of regular concrete deep beams was effectively improved by the addition of shear stirrups and that the adopted strengthening scheme improved both the shear strength and the deformation capacity of the beams. To forecast the shear capacity of reinforced-concrete deep beams, Prayogo et al. [4] introduced an AI method called OSVM-AEW, which combined two support vector machine models with the symbiotic organisms search algorithm. To coordinate the learning outputs, OSVM-AEW chooses the best possible SVM and LS-SVM parameters at the same time. In terms of the correlation coefficient, the determination coefficient, the mean absolute error, the mean absolute percentage error, and the root-mean-squared error, the experimental findings showed that OSVM-AEW performed exceptionally well. In this study, we discuss how OSVM-AEW could be used by structural engineers as a useful tool during the design phase of RC deep beams. The effects of section height, shear span to effective depth ratio, and compressive strength of concrete on the behaviour of diagonal cracking and crack width in RC deep beams were experimentally examined by Demir et al. [5]. We also carried out basic maths on shear reinforcement. The test findings demonstrated that shear strength improves with decreasing shear span to effective depth ratio and that load-carrying capacity grows with increasing section height. In addition, shear strength benefits greatly from confinement struts and compressive strength’s contribution to shear strength.

Hybrid concrete made from steel and polypropylene (PP) fibres combines the advantages of both types of fibre-reinforced concrete. Steel fibres improve the concrete’s strength and ductility, while polypropylene fibres make it more durable and crack-resistant. In hostile environments where corrosion is a key problem, steel-PP hybrid concrete showed encouraging results in increasing the mechanical characteristics and performance of concrete structures. Bridge decks, pavements, and earthquake-resistant structures are just some of the places one might see this tough stuff put to use. Steel-PP hybrid concrete is a ground-breaking innovation in the industry, with the potential to greatly enhance the functionality and durability of concrete buildings. Crack arresting in reinforced concrete was a topic investigated by Qian and Stroeven [6]. The total fibre content ranges from 0% to 0.95% and is made up of steel and polypropylene. The results indicate a beneficial synergistic effect for relatively modest displacements, but not for relatively large ones. The right hybrid fibre system can have a higher ultimate load-bearing capacity and fracture toughness than a mono-fibre system. Using four-point bending tests, Li et al. [7] described an experimental investigation of the flexural behaviour of steel-polypropylene hybrid fibre-reinforced concrete (HFRC). The research considered straight steel fibre, hooked-end steel fibre, and corrugated steel fibre, as well as monofilament polypropylene fibre. According to the findings, the flexural behaviour of HFRC improves when steel and polypropylene fibres are combined, with hooked-end fibre specimens faring the best. Concrete’s compressive, splitting tensile, and flexural strengths, as well as its post-peak ductility, were all enhanced by an increase in fibre volume fraction and steel fibre aspect ratio, while strength deterioration was mitigated. The mechanical reactions of steel-polypropylene hybrid fibre-reinforced concrete were simulated using a modified concrete damaged plasticity model (CDPM) described by Chi et al. [8]. (HFRC). The changes involve identifying parameters reliant on the fibre effect and discussing their impact on numerical findings. Independent experimental results validated the updated CDPM, demonstrating a high degree of agreement between numerical predictions and test outcomes. The strength and durability of steel fibre-reinforced concrete and steel-polypropylene hybrid fibre-reinforced concrete were compared by Song et al. [9]. Drop-weight tests were performed on 48 discs to determine their strengths. While there was less variety in the strengths of the hybrid concrete, there was more variation in the percentage increase in blows after the first crack appeared. Statistical analysis demonstrated that both types of concrete deviated significantly from normal distributions. In comparison to steel fibre-reinforced concrete, hybrid concrete offered less of an increase in strength and a smaller percentage increase. The first crack and failure strengths were somewhat more consistent in the hybrid concrete.

In structural engineering, the concrete damage plasticity (CDP) model is a popular numerical model for evaluating the response of concrete structures to different loads. Concrete’s complicated behaviour, including cracking, crushing, and plastic deformation, are all accounted for in the CDP model, which is a continuum-based model. The post-peak behaviour of concrete may be captured by this non-linear model, making it a vital tool for the design and analysis of structures subjected to severe stresses. The CDP model was effectively implemented in several contexts, such as earthquake engineering, blast-resistant building design, and reinforced concrete construction. particularly in earthquake-prone places where ductility is of crucial concern, this finding may have significant consequences for the design and construction of reinforced concrete structures. The concrete damage plasticity (CDP) model was first proposed by Lubliner et al. [10], and it is a continuum model for concrete that can reproduce the results of cracking, crushing, and plastic deformation. The research provided a theoretical foundation for the CDP model and provided various case studies of its use in analysing the behaviour of concrete structures. Nonlinear analysis of reinforced concrete structures, which takes into account inelastic, time-dependent, and cracking, is gaining popularity, as Hafezolghorani [11] explained. To simulate the behaviour of unrestrained concrete, which can be described in tabular forms for different concrete grades, a simplified concrete damage plasticity (SCDP) model was developed. Damage parameters, strain hardening/softening criteria, and other model parameters were tabulated for easy reference. The study used the finite element method to assess a simply supported prestressed beam made of four different concrete grades, and the results correlated well with prior works and empirical formulations. Finite element modelling can be used to accurately represent the behaviour of reinforced concrete beams, as demonstrated by Sumer and Aktas [12]. For an accurate simulation of reinforced concrete’s load-deflection behaviour, it is important to couple elastic damage models with elastic–plastic constitutive laws. The research provided an equation for the damage parameter and recommended modelling methodologies for testing the model’s sensitivity to changes in mesh density, dilation angle, and concrete fracture energy. Three independent experimental checks supported the models’ accuracy. To model the behaviour of high-strength concrete under static and dynamic loading conditions, Minh et al. [13] introduced a new concrete damage plasticity model. The model is an updated version of the original CDP model implemented in ABAQUS. Compressive and tensile stress–strain curves were modelled with mesh size taken into account. Predicting and modelling concrete structural elements using damage plasticity models was studied by Al-Zuhairi et al. [14]. The research concluded that an analytical model based on stiffness deterioration was superior to previous methods for predicting damage in concrete structures subjected to compressive loads.

The use of the CDP (concrete damaged plasticity) model in modelling fibre hybrid concrete is a suitable approach to capture the behaviour of such materials. The CDP model is widely used in simulating the response of concrete structures, including fibre-reinforced concrete. Several studies took into consideration modelling fibres inside concrete where having an experimental study will lead to high cost and time consumption. The behaviour of hybrid concrete is reflected by presenting CDP parameters that were changed to fit the required compression and tension behaviour [15,16,17,18,19].

Ibrahim and Rad [20] used a probabilistic design that considered random concrete and CFRP parameters, as well as complimentary strain energy values, to study the plastic behaviour of haunched beams strengthened with carbon fibre polymers CFRP. Models with restricted plasticity can be created using the reliability index as a constraining parameter. The outcomes demonstrate that randomness influences the models’ behaviour, altering the values of load and deflection and the proportion of damaged tension concrete. The reliability index is a limiting index that factors in the randomness of both concrete qualities and complementary strain energy. Damage severity and failure likelihood both rise in tandem with the output load.

This study aims to investigate the effect of using different mixes of steel fibres (SF) and polypropylene fibres (PPF) on the behaviour of simply supported reinforced concrete deep beams, both experimentally and numerically. The study aims to determine the most effective percentage of fibres for improving the strength and ductility of the deep beams. The novelty of the study was highlighted in the investigation of hybrid polymer reinforced concrete (HPRC) and the use of a numerical calibration process. The numerical process was held using the concrete damage plasticity (CDP) model in ABAQUS to investigate the behaviour of HPRC deep beams with different material combinations based on the experimental results. Firstly, two reinforced deep beams were tested experimentally, one with normal concrete (0%SF–0%PPF) and the other with (75%SF–25%PPF). The test results were then compared to study the effect of fibre existence. Additionally, six concrete mixes were prepared and tested to obtain the concrete parameters for each, with different percentages of SF and PPF, including (0%SF–0%PPF), (1%SF–0%PPF), (0.75%SF–0.25%PPF), (0.5%SF–0.5%PPF), (0.25%SF–0.75%PPF), and (0%SF–1%PPF). Numerical models were then calibrated using ABAQUS based on the experimental results, and calibrated CDM parameters were used to match the experimental concrete behaviour. The calibrated models were then compared, taking into account ultimate load, deflection, cracking patterns, and concrete damage in tension and compression, to investigate the most effective percentage of fibres for improving the strength and ductility of the deep beams. The study aimed to explore the potential of hybrid polymer-reinforced concrete (HPRC) to enhance the strength and ductility of reinforced concrete structures using a combination of steel and polypropylene fibres. The recent studies did not cover the behaviour of deep beams with these ranges of fibres percentages; also, the presented comparisons showed the compression and tension behaviour of the tested mixes in details by presenting stress–strain relationships. Moreover, this study showed the significance of having hybrid concrete on the concrete and steel damages using ABAQUS.

This study’s significance lies in the insights it provides for improving fibre-reinforced polymer composites for building applications. The research shed light on the performance of HPRC in reinforced concrete deep beams and how it might be improved. Engineers and researchers can use the results as a guide when deciding which fibre combinations and percentages to use to achieve specific structural goals. There are several highlighting features of this study that make it a novel study. It begins by investigating deep beams experimentally, a specific structural feature that plays a significant role in many different building projects and where hybrid polymer concrete can be useful. To further our comprehension of how various fibre combinations affect the behaviour of deep beams, this study also examined the combined impacts of SF and PPF, taking varying percentages into account; these percentages were derived by testing several specimens and then, their tension and compression behaviours were investigated. Finally, experimental testing and numerical analysis were utilised in this study for a more thorough evaluation and validation of the findings, which increases the study’s credibility and practicality.

After this introduction, Section 2 includes the details of the constitutive model of concrete. Then, Section 3 presents the experimental work that includes the model details, test set-up and the discussion of the experimental results. Furthermore, Section 4 illustrates the numerical model details, the calibration process and the discussion of the numerical results. Finally, the most important conclusions and the study summary are summarized in Section 5.

## 2. Constitutive Model of Concrete

The concrete damage plasticity (CDP) model is a constitutive model used to simulate the nonlinear behaviour of concrete subjected to loading, such as compression, tension, and bending. It is based on the concept of continuum damage mechanics, which considers that the material undergoes damage and degrades as it is subjected to loading. It considers two different types of damage: tensile and compressive. In the tensile damage, the model assumes that the material undergoes microcracking, which leads to a reduction in the material’s stiffness and strength. In compressive damage, the model assumes that the material undergoes crushing, which leads to a reduction in the material’s stiffness and strength. It also considers the effects of plasticity, which means that the material undergoes permanent deformation even after the load is removed. The model uses an isotropic hardening rule to capture the plastic behaviour of the material. To implement the CDP model, a set of parameters is required to describe the material’s behaviour. These parameters include tensile and compressive strengths, Young’s modulus, and others which are typically obtained through experimental testing.

Detailed explanations of the procedure can be found in the relevant publications and monographs. In this article, a quick overview of the concrete constitutive model is provided and then extended in a few key ways. The entire strain tensor value $\epsilon_{ij}$ can be broken down into an elastic component $\left(\epsilon_{ij}^{el}\right)$ and a plastic component $\left(\epsilon_{ij}^{pl}\right)$ using the Prandtl–Reuss theory in conjunction with elasto-plastic deformations, as explained in more detail below [10,21,22,23]:(1)$$\epsilon_{ij}=\epsilon_{ij}^{el}+\epsilon_{ij}^{pl}.$$

The scalar damage elasticity equation also gives us insight into the interactions between forces and strains within a material:(2)$${\hat{\sigma }}_{ij}=D_{ijkl}^{el}*\left(\epsilon_{ij}-\epsilon_{ij}^{pl}\right)$$
therefore, $D_{ijkl}^{el}$ is abbreviation for decreased elastic stiffness.
(3)$$D_{ijkl}^{el}=\left(1-d\right)D_{0}^{el}$$
where $D_{0}^{el}$ is the undamaged material’s initial elastic stiffness and d is a scalar degradation variable with values between 0 (undamaged) and 1 (completely damaged). According to the scalar-damage hypothesis, there is just one degradation variable, denoted by the variable d, that accounts for the isotropic decrease in stiffness. The effective internal force is described using common ideas from continuum damage mechanics, which are as follows:(4)$${\bar{\sigma }}_{ij}=D_{0}^{el}*\left(\epsilon_{ij}-\epsilon_{ij}^{pl}\right).$$

The scalar reduction relation relates the internal force ${\hat{\sigma }}_{ij}$ to the effective internal force $\;{\bar{\sigma }}_{ij}$:(5)$${\hat{\sigma }}_{ij}=\left(1-d\right)\cdot {\bar{\sigma }}_{ij}$$

The effective internal force ${\bar{\sigma }}_{ij}$ is equal to the internal force ${\hat{\sigma }}_{ij}\;$ when d = 0. As the region of the effective internal force is what resists the external loads, it becomes more representative than the internal force when damage occurs. The nominal stress and the decreased elastic tensor described in Equation (4) can be incorporated into a new form of Equation (2), creating the following equation:(6)$${\hat{\sigma }}_{ij}=\left(1-d\right)D_{0}^{el}*\left(\epsilon_{ij}-\epsilon_{ij}^{pl}\right).$$

This internal force–strain relationship forms the constitutive model of damage plasticity:(7)$${\hat{\sigma }}_{ij}=\left(1-d\right)\cdot {\bar{\sigma }}_{ij}\to {\hat{\sigma }}_{ij}=\left(1-d_{t}\right){\bar{\sigma }}_{t_{ij}}+\left(1-d_{c}\right){\bar{\sigma }}_{C_{ij}}$$
where $d_{c}$ and $d_{t}$ represent compression and tension damage, respectively, and range from 0 (the undamaged case) to 1 (the fully damaged case), and ${\bar{\sigma }}_{t}$ and ${\bar{\sigma }}_{c}$ represent the effective internal force under tension and compression, respectively. Compressive crushing and tensile cracking were typically accounted for in damage models for concrete. However, due to the intricacy of the degrading mechanism exhibited by the uniaxial cyclic behaviour of concrete, the uniaxial reaction of concrete was investigated (opening and closing of the created micro-cracks). As illustrated in Figure 1, it is expected that plasticity damage influences the uniaxial compressive and tensile response of concrete.

Concrete’s uniaxial compressive and tensile reaction to compression and tension stress, as described by the concrete damage plasticity model, is shown by:(8)$$\sigma_{t}=\left(1-d_{t}\right)E_{0}\left(\epsilon_{t}-\epsilon_{t}^{pl,h}\right)$$
(9)$$\sigma_{c}=\left(1-d_{c}\right)E_{0}\left(\epsilon_{c}-\epsilon_{c}^{pl,h}\right)$$
where $E_{0}$ represents the Young’s modulus of the undamaged material and $\epsilon_{t}^{pl,h}$ and $\epsilon_{c}^{pl,h}$ represent the comparable plastic strains in tension and compression. This allows us to derive the effective uniaxial compressive $\bar{\sigma_{t}}$ and tensile ${\bar{\sigma }}_{c}$ stresses:(10)$$\bar{\sigma_{t}}=\frac{\sigma_{t}}{\left(1-d_{t}\right)}=E_{0}\left(\epsilon_{t}-\epsilon_{t}^{pl,h}\right)$$
(11)$${\bar{\sigma }}_{c}=\frac{\sigma_{C}}{\left(1-d_{c}\right)}=E_{0}\left(\epsilon_{c}-\epsilon_{c}^{pl,h}\right)$$
where tensile strain $\epsilon_{t}$ equals $\epsilon_{t}^{pl,h}+\epsilon_{t}^{el}$, and compressive strain $\epsilon_{c}$ equals $\epsilon_{c}^{pl,h}+\epsilon_{c}^{el}$. Hence, $\epsilon_{t}^{el}$ and $\epsilon_{c}^{el}$ are the equivalent elastic strains in tension and compression, respectively.

Using the expressions: $\epsilon_{t}$ = $\epsilon_{t}^{pl,h}+\epsilon_{t}^{el}$ and $\epsilon_{c}$ = $\epsilon_{c}^{pl,h}+\epsilon_{c}^{el}$ for tensile and compressive strains, respectively. As a result, the elastic strains under tension and compression are denoted by $\epsilon_{t}^{el}$ and $\;\epsilon_{c}^{el}$, respectively.

In conclusion, the calibrated model represents tension and compression concrete behaviour using the CDP constitutive model, whereas these two behaviours are shown utilising the concrete’s properties in tension and compression according to Figure 1, and then, these properties were evaluated within ABAQUS in order to validate our numerical model. After running the ABAQUS, the CDP parameters were achieved and our model would be ready for the test of the different concrete mixes. The parameters that were considered to be calibrated in this study included:Dilatation angle: The dilatation angle is a parameter that characterizes the volumetric change in a material during plastic deformation. It represents the angle between the direction of the maximum principal stress and the direction of maximum volume change. In Abaqus CDP, the dilatation angle is used to control the volumetric response of the material, particularly in compression.Eccentricity: The eccentricity parameter is used to describe the anisotropic behaviour of a material during plastic deformation. It represents the deviation of the plastic strain from the direction of maximum shear stress and can be used to control the material’s yield surface shape in the stress space.fb0/fc0: fb0 and fc0 are parameters used to describe the initial compressive and tensile strength of concrete in CDP models. They are used to control the initial stiffness and strength of the material and are typically determined experimentally.K: The isotropic hardening parameter “k” is used to describe the hardening behaviour of a material during plastic deformation. It determines the rate at which the yield stress increases with plastic strain or plastic strain rate and is an important parameter in accurately simulating the material’s plastic behaviour under various loading conditions.Viscosity parameter: The viscosity parameter is used to describe the material’s viscous behaviour, which is the ability of a material to resist deformation over time when subjected to a constant load. In CDP models, the viscosity parameter can be used to control the rate at which the material’s yield stress increases with increasing strain rate, or to simulate the material’s creep behaviour.

## 3. Experimental Work

The behaviour of a deep beam reinforced with steel and polypropylene fibres was studied through experiments conducted at the University of Technology’s lab. The experiments involved the construction and testing of a pair of deep, simply supported beams subjected to symmetric two-point loading. The overall height (h) of the two beams was 400 mm, the width (b) was 150 mm, the total length (L) was 1400 mm, and the clear span (Ln) was 1070 mm. The Ln/h ratio was equal to 2.675, which was less than the 4 that is recommended by the provision of the ACI 318M-14 code [24]. The primary longitudinal tension reinforcement in each specimen was made up of three steel bars with a nominal diameter of 16 mm. Additionally, as shown in Figure 2, both the vertical and horizontal webs were reinforced by steel bars of 4 mm in diameter, spaced at 60 mm C/C to meet the minimum spacing required by ACI 318M- 14 [24]. Normal concrete was used to form one beam, whereas steel-polypropylene fibre concrete was used to cast the other with 0.25%SF and 0.75%PPF fibre percentages.

As can be seen in Figure 3, DRACO^®^ Company’s hooked end SF class C, type A was used throughout the experimental procedure. In addition, the manufacturer provided data in Table 1 describe SF’s technical characteristics. Nonetheless, as can be seen in Figure 4, high-performance monofilament polypropylene fibres were utilised here. Table 2 showed the typical PPF features (aspect ratio L/D = 667) in use.

Furthermore, deformed longitudinal steel bars with a nominal diameter of 16 mm were used in this investigation, and deformed steel with a diameter of 4 mm was used for both vertical and horizontal shear reinforcement. The characteristics of the steel bars used in this experiment are shown in Table 3. Three specimens of each diameter were tested for tensile strength in accordance with ASTM A370-14 [25]. In addition, the concrete properties for the six different mixes were characterised by testing a total of 126 control specimens, including 90 cylinders of 200mm × 100 mm, 12 cylinders of 300mm × 150mm, and 24 prisms of 100mm × 100mm × 400mm (Figure 5). Test results for various concrete properties are shown along with data about these six mixtures in Table 4; the presented percentages of fibres were chosen to cover almost all the probability ranges from no fibre at all until having 100 percent.

Steel-polypropylene fibres added to concrete might have both positive and negative results depending on the specific circumstances. Adding steel fibres to concrete is a common practice for increasing the material’s strength and durability. Steel fibres enhance concrete’s tensile strength and toughness, which helps prevent cracking and boosts the material’s overall performance. On the other side, polypropylene fibres are frequently utilised to limit concrete’s potential to crack. They do not do much to boost the concrete’s strength, but they can keep it from shrinking and splitting as much. Polypropylene fibres are sometimes added to concrete mixes to improve durability and reduce the potential for cracking, both of which can help to maintain the compressive strength of the concrete over time, despite the fact that they can reduce the stiffness of the concrete and, thus, lead to a lower modulus of elasticity. However, the concrete’s compressive strength can be reduced if an excessive number of polypropylene fibres is added during the mixing process. This is due to the fact that the fibres may form air pockets in the concrete, thus weakening the material.

Adding steel fibres to concrete increases its tensile strength and makes it less likely to crack in tension. As can be seen in Table 4, although increasing PPF decrease most of the concrete parameters if compared with the effect of adding SF; however, the parameters are mostly enhanced if compared with the normal concrete without any fibres added. The influence of steel fibres and polypropylene on the mode of failure is represented in Figure 6; although both specimens failed, the one made of hybrid concrete appeared to fail in a more coherent failure. The combination of steel and polypropylene fibres in a concrete mix can provide a more comprehensive reinforcement strategy that enhances the performance and durability of the concrete in a variety of applications. By working together, steel and polypropylene fibres can help to improve the overall strength, toughness, and resistance to cracking and deformation of the concrete.

The beams that were tested experimentally in this study had the properties shown in Figure 2; the first beam (M1B) had the M1 mix properties presented in Table 4, where no fibres were used. On the other hand, the second tested beam (M3B) had the presented M3 mix properties shown in Table 4, where a percentage of 25% PPF and 75% SF was used. Tests conducted produced data, as observed in the experiments, that are illustrated as cracking patterns and the relationship between load and deflection as depicted in Figure 7 and Figure 8. Both types of beams examined failed by shear failure, with hybrid concrete beams performing much better than normal concrete ones. Both standard and hybrid concrete deep beams failed in the same way, with the concrete splitting along the line between the edge of loading and the supporting points and the concrete being crushed at the edge bearings at the load positions for the case of the normal concrete beam. Hybrid beams tended to have fewer cracks overall, especially in the flexural direction, when the fibre was included in the concrete. The presence of steel fibre and polypropylene increased the ultimate load value brought by hybridization from 800 kN to 900 kN. The presence of fibres in the concrete deep beam lowered the deflection of the hybrid concrete deep beam as compared to the matching normal concrete deep beam as the fibres raised the stiffness of the hybrid concrete.

Generally, it can be seen that the type of the crack spread was different for the two tested beams, the crack that caused the failure in beam M1B seemed to be wide and straight, accompanied by compression crushing of concrete near the loading area; on the other hand, the beam with fibres seemed to have a different crack initiating method, where the crack that caused the failure consisted of small tight cracks and no compression cracks are presented. In this, the effect of the fibres occurred in tightening the cracks and increasing the ultimate load, as presented in Figure 8.

## 4. Numerical Modelling

In order to further investigate the effect of having different fibre percentages, this section considered validating the two tested deep beams using ABAQUS. The hybrid concrete beam M3B was validated using the concrete damage plasticity (CDP) model by having a calibration process in order to obtain the parameters that reflect the existence of the effect of the fibres on the beam behaviour. Starting with defining the material properties of the hybrid concrete, such as elastic modulus, Poisson’s ratio and material strength and damage parameters. Then, calibrating the CDP model for the hybrid concrete using experimental data, such as the stress–strain curves or the test results. Finally, fitting the damage parameters to the experimental data to calibrate the model to the actual behaviour of the hybrid concrete beams. Moreover, the six mixes presented in Table 4 were considered to be provided in tension and compression forms as shown in Figure 9. Observing this figure, it can be seen that in the compression behaviour, the maximum steel fibre percentages (M2) gave the highest compression stress; however, in the softening part (pre the peak), it had the lowest values. Then, the curves degraded into lower compression stress as the SF ratio decreased. On the other hand, the tension behaviour was not so organized, where it can be noticed that all the mixes with fibres produced better tension stress values than the normal concrete.

The 3D finite element models were developed using data from laboratory experiments. Using the CDP model to represent concrete within ABAQUS [26], six simply supported deep beams were modelled based on experimental results from the two tested beams and the six tested hybrid concrete mixes, as seen in Figure 10, where Poisson’s ratio of concrete v = 0.2, and measurements and explanations are provided for concrete’s compressive and tensile performance during adoption testing. To simulate the nonlinear behaviour of the beams using the damage plasticity model, a finite element failure analysis was conducted. After performing sensitivity analyses, the CDP plasticity parameters were assumed as given in Table 5 for both normal concrete and hybrid concrete. Each one of these parameters were tested considering specific ranges where the presented values were the values that reflected the experimental behaviour accurately. The concrete damage plasticity data included compressive crushing and tensile cracking as failure mechanisms of the material taken from mechanical properties tests of specimens. The CDP parameters calibration process included changing these parameters in different ranges to acquire the best match that reflects the hybrid concrete behaviour, it was realized that dilatation angle and viscosity parameters were the most effective characteristics that affected the results mainly, while the other parameters had no or slight effect on the results.

The finite element model was constructed from concrete and reinforcement materials, with the concrete represented by a solid element with eight nodes (C3D8: eight-node first-order hexahedral element with an exact numerical integration) and the reinforcement bars represented by beam elements with a 2-node linear beam in space (B31: Timoshenko beam). An embedded zone was used to mimic the link between the longitudinal and transverse reinforcements and the concrete, while the boundaries were set so that they would be equivalent to those observed in a laboratory setting. In addition, identical experimental circumstances were achieved by applying a vertical concentrated load at each point load of the beam and then distributing the loads via the coupling effect. Both numerical accuracy and computation time can be affected by the mesh size, and so, a size study was performed to see how different mesh sizes performed and to choose the most suitable size. The optimal mesh size was then applied to provide a precise result, with the total number of elements equal to 25,110 for the beams.

The numerical models calibrated with the CDP model in ABAQUS showed good agreement with the experimental results, validating the use of the model for predicting the behaviour of HPRC deep beams as shown in Figure 11 and Figure 12. Figure 11 depicts the similarities between the concrete damage patterns (tension damage) observed between the numerical and experimental data, it could be noticed that the use of fibres limited the damaged red areas in the flexural part of the beam greatly. Where the blue colour indicates non-affected regions ($d_{t}=0$) and red the colour indicates severely damaged regions ($d_{t}=1$), so that the level of damage is represented graphically. As can be observed in Figure 12, the two methods (experimental and numerical) produced similar results.

The addition of different percentages of steel fibres (SF) and polypropylene fibres (PPF) to concrete had various effects on the crack pattern, load capacity, deflection, and damage of reinforced concrete beams. After the validation process, the six beams were modelled and tested numerically, revealing interesting results. Load deflection curves are presented in Figure 13, while the damage patterns of concrete and reinforcement steel are presented in Table 6. Here are some general reflections for each beam:M1B (0%SF–0%PPF): This is the fibre-free control beam. It has the lowest loading capacity where the absence of fibre reinforcement makes the beam more subject to fracture under stress. In comparison to fibre-reinforced beams, the damaged area seems more numerous (more red areas). Moreover, in this situation, the beam is more subject to deflection under load, increasing sagging and deformation. Therefore, beam M1B has the lowest load capacity and the highest deflection values when compared to the other mixtures even though the M1 mix has a better concrete compression strength than M3, M4, M5, and M6, but the absence of fibres was reflected by the general behaviour of the beam. Moreover, reinforcement steel bars undergo high stresses especially in the flexural area and along the major shear-damaged area.M2B (1%SF–0%PPF): Adding 1% SF to concrete improves its tensile and flexural strength and enhances its resistance to cracking. However, it may also lead to an increase in brittleness. It leads to smaller and less frequent cracks compared to the control beam without fibres. The flexural damage that appeared may be narrower and more tightly spaced. However, it does not cover the whole part, as was observed in the M1B model. The addition of SF helped to reduce deflection by providing additional stiffness and strength to the beam. The addition of 1% SF increased the load capacity of the beam remarkably, due to the additional tensile strength provided by the fibres. This mix has enhancement in the general behaviour of the beam if compared with the M1B case, especially in the flexural part; however, it was not the best tested case. In addition, it can be seen that the longitudinal reinforcement bars in the mid span flexural area are not damaged even after the failure, which reflects the steel fibres’ effectivity in tension strength.M3B (0.75%SF–0.25%PPF): Adding 0.25% PPF to the mix in addition to 0.75% SF can further improve the toughness and ductility of the material compared to the mix with 1% SF only. The addition of 0.25% PPF can affect the brittleness of the concrete. The damage that appeared in the middle of the beam seems a bit higher than in the previous case, however, it is still better if compared with the M1B case. The combined effect of SF and PPF improved the stiffness and resistance to deflection of the beam, as the PPF helps to absorb energy and reduce the risk of brittle failure. That is why M3B gave the best results in terms of load capacity between all the tested models. It can be seen also that the reinforcement steel damage was not too severe if compared with the M1B model.M4B (0.5%SF–0.5%PPF): Using equal percentages of SF and PPF can balance the properties of the concrete and provide improved flexural strength, ductility, and resistance to cracking and fatigue. The combination of SF and PPF can lead to a more balanced distribution of cracks across the beam, with smaller and less frequent cracks compared to the control beam (without fibres). The balanced combination of SF and PPF can provide a further improvement in load capacity, as the fibres work together to resist the applied loads if compared with the M1 mix. However, this improvement in the load capacity is not the best if compared with the M3 mix, where it can be seen that the results curves from now on for the next beams will have less loading capacity but less deflection values also.M5B (0.25%SF–0.75%PPF): Using a higher percentage of PPF (0.75%) with a lower percentage of SF (0.25%) can increase the toughness and energy absorption of the material. The higher percentage of PPF produced smaller deflection values, with a more occurrence of the damaged area in the mid-span of the beam. However, reducing the SF and using higher PPF weakened the general behaviour of the beam, where it can be seen that the longitudinal reinforcement started to present some damaged areas.M6B (0%SF–1%PPF): Adding 1% PPF to the mix can enhance the durability of the material and improve its resistance to shrinkage cracking, as well as provide some enhancement to its flexural strength. Beam M6B has a better load capacity than M1B, but it comes last if compared with the other hybrid beams even though it has the best enhanced and reduced deflection value among all the six mixes. Additionally, in this model, the damaged areas in the shear part happen to be less intensive if compared with all other cases, but it has the highest damaged flexural area among the hybrid models. On the other hand, because of the absence of steel fibres, the longitudinal reinforcement in this case is almost totally damaged.

The effect of different percentages of steel fibres (SF) and polypropylene fibres (PPF) on the behaviour of concrete beams was evaluated in terms of their impact on crack pattern, load-carrying capacity, deflection, and steel and concrete damage. In summary, the addition of fibres to concrete can significantly affect the crack pattern and the damage behaviour of concrete beams. It also improves the load-carrying capacity of the beam by enhancing its strength and reduces the deflection, where fibres existence affect the deflection of the beams by changing its stiffness and ductility. In general, the addition of fibres can increase the beam’s stiffness, which can reduce the amount of deflection under loading. However, the effect of the fibre content on the load-carrying capacity can be complex, as an increase in fibre content can also lead to an increase in brittleness, which can reduce the overall strength of the material; this occurred in the numerical models where increasing PPF percentage weakened the beams but not using any PPF percentages did not give the best behaviour among the tested models. Regarding steel and concrete damage, the use of fibres can also affect the damage behaviour of the steel and concrete in the beam. SF can improve the behaviour of steel bars and concrete in flexural areas. On the other hand, PPF can reduce the damage a bit in the shear area but it has almost lighter effect on the flexural parts.

## 5. Conclusions

By investigating the effect of different mixes of SF and PPF, this study aimed to contribute to the knowledge base on HPRC and its potential to improve the performance of reinforced concrete deep beams.

The experimental results showed that the addition of SF and PPF could significantly enhance the load-bearing capacity of reinforced concrete deep beams. The presence of fibres also affected the ductility of the beams, as evidenced by the decrease in deflection before failure. The cracking patterns of the beams with fibres were also more cohesive, uniform and narrow compared to the beams without fibres.

The numerical models calibrated with the CDP model in ABAQUS showed good agreement with the experimental results. The results showed that the optimal percentage of fibres depends on the desired performance. For instance, if the goal is to improve the load-bearing capacity and crack distribution, then a 0.75%SF–0.25%PPF combination is preferred. Conversely, if the aim is to improve the deflection, i.e., producing lower deflection values, then PPF higher content is preferable.

In conclusion, the addition of steel fibres (SF) and polypropylene fibres (PPF) to concrete has varying effects on the crack pattern, load capacity, deflection, and damage of reinforced concrete beams. The control beam without fibres (M1B) has the lowest load capacity and highest deflection values compared to the other mixes. The addition of 1% SF (M2B) improved the tensile and flexural strength of the beam, reduced deflection, and increased load capacity. The addition of 0.25% PPF with 0.75% SF (M3B) improved toughness and ductility, reduced brittleness, and provided the best results in terms of load capacity. Using equal percentages of SF and PPF (M4B) balanced the properties of the concrete and provided improved flexural strength. Using a higher percentage of PPF with a lower percentage of SF (M5B) weakened the general behaviour of the beam. Adding 1% PPF (M6B) provided some enhancement to flexural strength if compared with the normal concrete but resulted in almost total damage to the longitudinal reinforcement due to the absence of steel fibres. Overall, the addition of SF and PPF can improve the properties of reinforced concrete beams, but the combination and percentage of fibres used should be carefully considered to achieve optimal results.

The following are some ideas for further research based on the results of this study:Finding the optimized ratio of steel fibres (SF) to polypropylene fibres (PPF) for various performance criteria is a potential area for future study;Although SF and PPF were the primary focus of this study, other fibre types such as carbon fibres and glass fibres exist and can be considered;Long-term impacts of fibre addition on the behaviour of reinforced concrete beams were not the primary focus of the present study. The long-term effects of these fibres, such as their resistance to moisture, temperature changes, and chemical exposure, can be studied in future studies;Although deep beams were the focus of this study, columns, slabs, and walls are all potential targets for future research into the effects of SF and PPF.

## Figures and Tables

**Figure 1 polymers-15-02340-f001:**
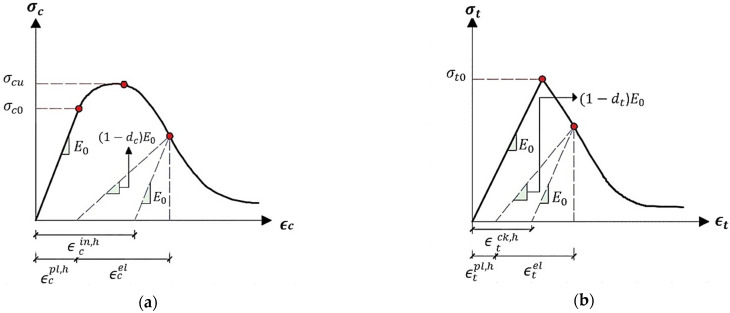
Concrete response to uniaxial loading condition: (**a**) Compression, (**b**) Tension.

**Figure 2 polymers-15-02340-f002:**
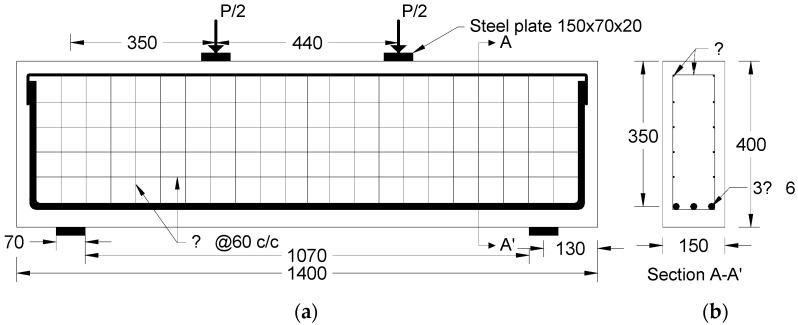
Deep beam geometry (all dimension are in mm): (**a**) front view, (**b**) cross-section.

**Figure 3 polymers-15-02340-f003:**
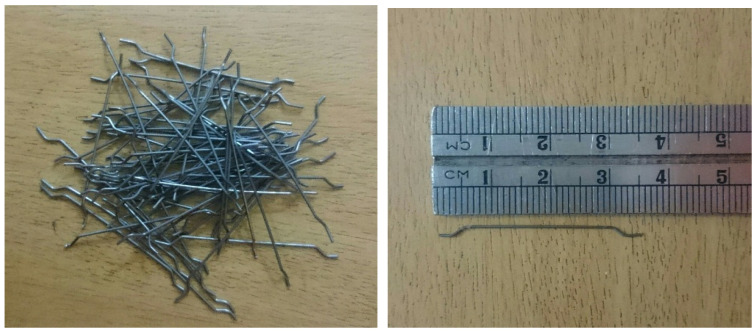
Hooked End Macro Steel Fibres.

**Figure 4 polymers-15-02340-f004:**
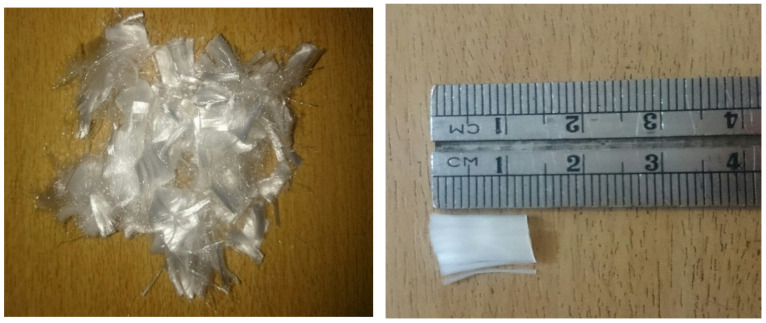
Monofilament Polypropylene Fibres.

**Figure 5 polymers-15-02340-f005:**
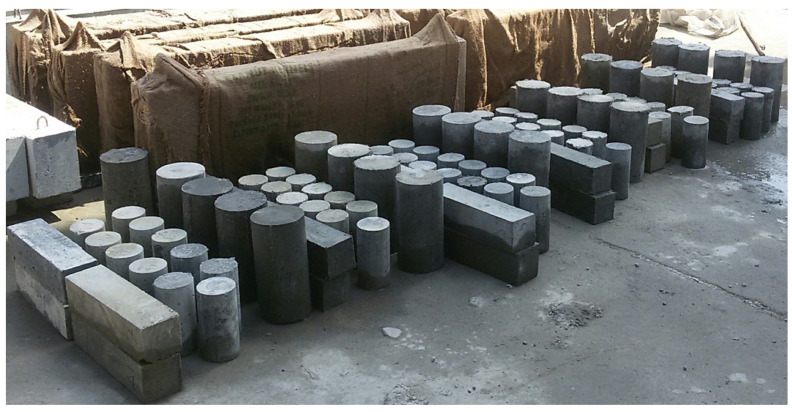
The total 126 tested control specimens.

**Figure 6 polymers-15-02340-f006:**
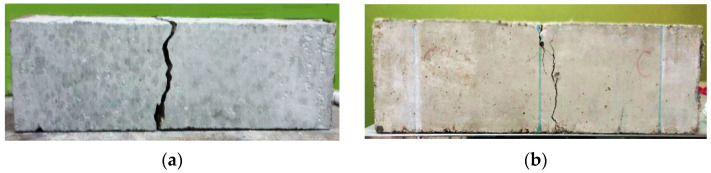
Mode failure of prisms: (**a**) Without fibre, (**b**) With fibre.

**Figure 7 polymers-15-02340-f007:**
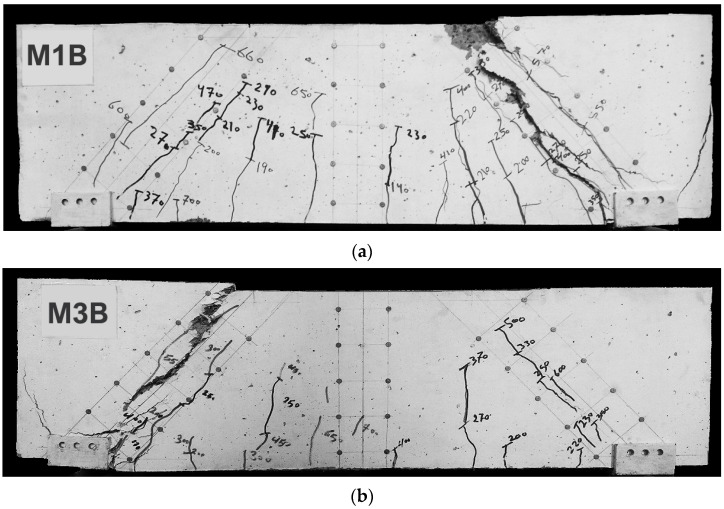
Crack pattern and failure shape for the tested beams: (**a**) M1B, (**b**) M3B.

**Figure 8 polymers-15-02340-f008:**
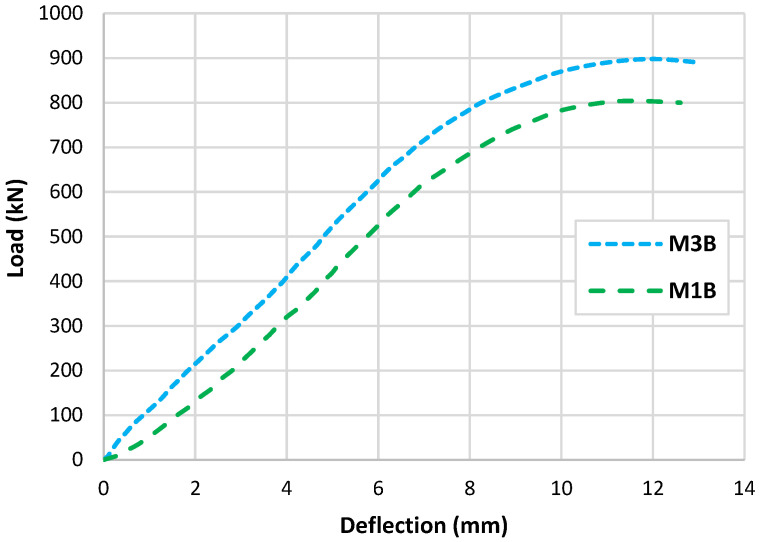
Experimental load-deflection relationships.

**Figure 9 polymers-15-02340-f009:**
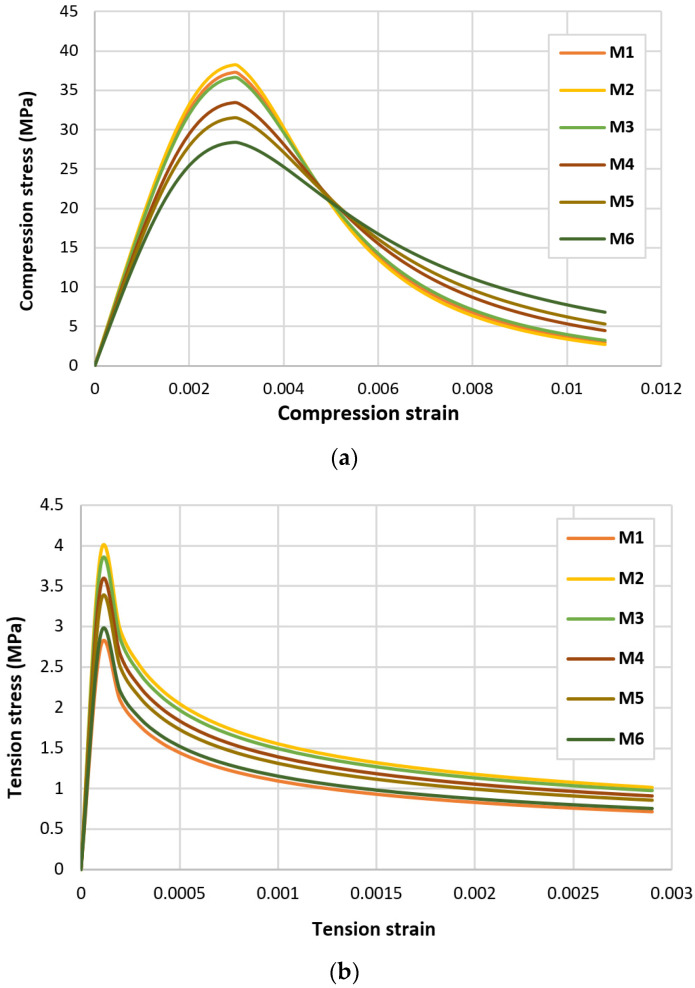
Behaviour for the six mixes: (**a**) In compression, (**b**) In tension.

**Figure 10 polymers-15-02340-f010:**
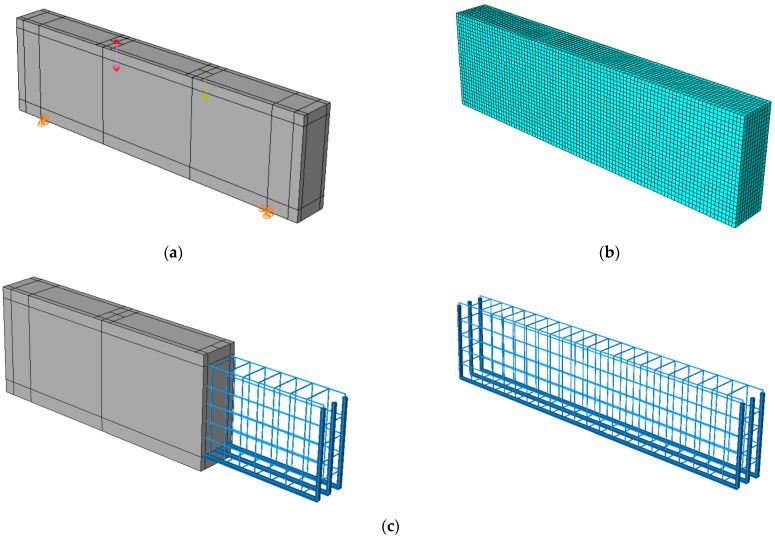
Numerical model details: (**a**) loading and supporting conditions, (**b**) mesh details, (**c**) reinforcement details.

**Figure 11 polymers-15-02340-f011:**
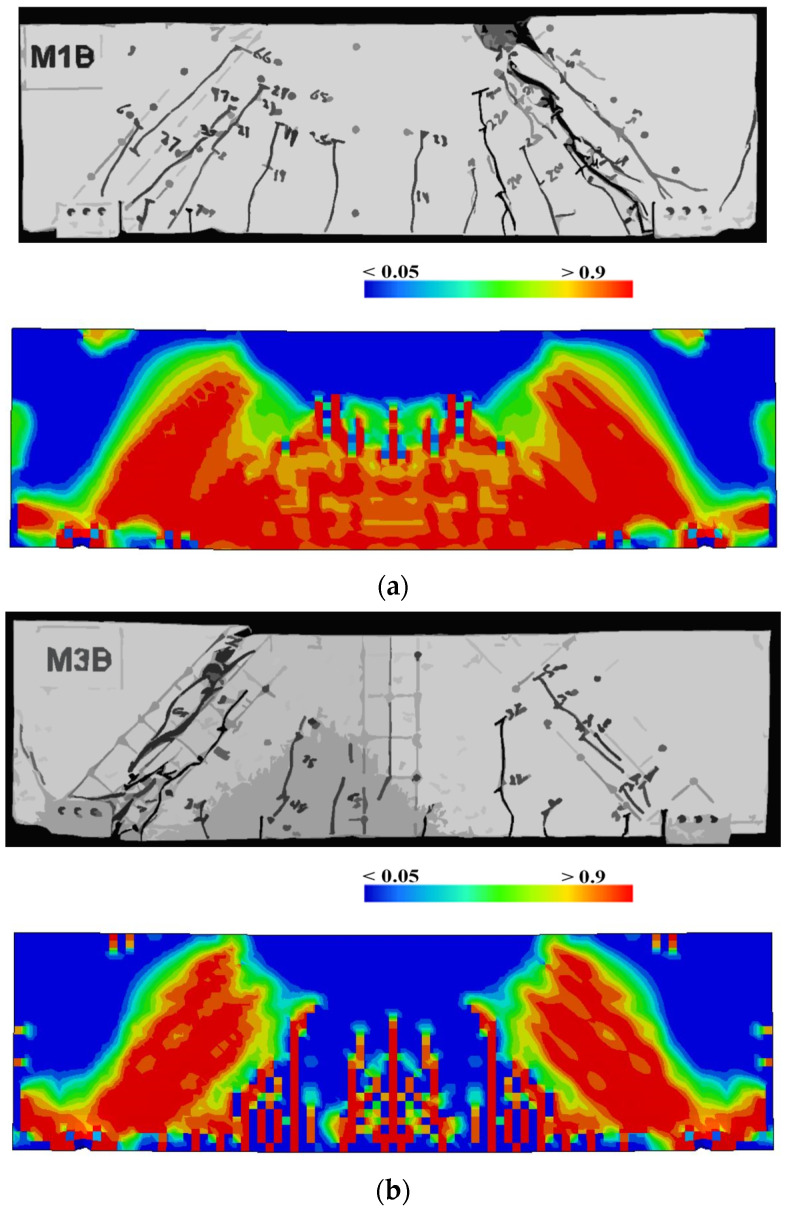
Comparison of damage patterns: (**a**) M1B beam, (**b**) M3B beam.

**Figure 12 polymers-15-02340-f012:**
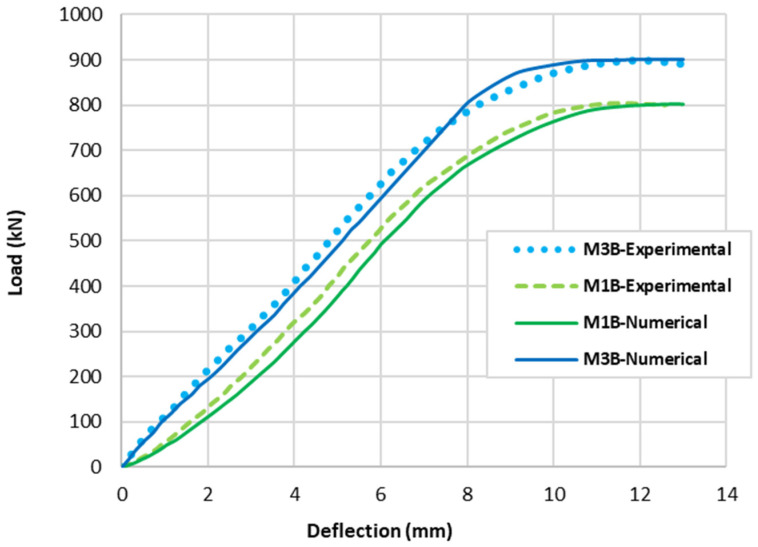
Numerical model validation.

**Figure 13 polymers-15-02340-f013:**
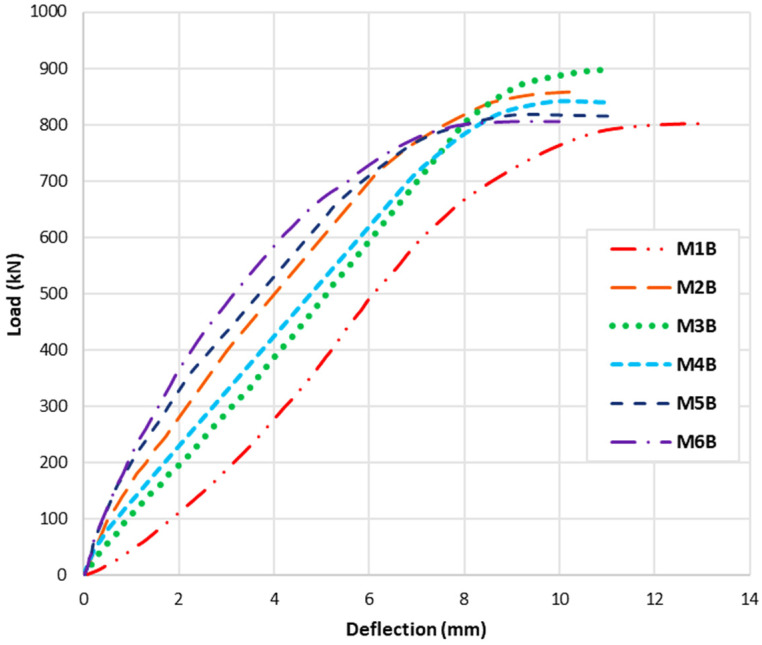
Numerical load-deflection relationships for all six models.

**Table 1 polymers-15-02340-t001:** The technical properties of Steel Fibres.

Properties	Result
Length L (mm)	35
Diameter (mm)	0.55
Aspect ratio (L/D)	64
Wire tensile strength (MPa)	~1100
Wire bending strength (MPa)	~800
Elongation at break	<2%

**Table 2 polymers-15-02340-t002:** Properties of the used PPF.

Properties	Result
Length L (mm)	12
Diameter D (mm)	0.15
Aspect ratio (L/D)	80
Tensile strength (MPa)	1300
Density (kg/m^3^)	900
Cross Sectional area	Circular

**Table 3 polymers-15-02340-t003:** Tensile test results of steel reinforcement.

Bar Diameter (mm)	Yield Strength (MPa)	Tensile Strength (MPa)
16	595	680
4	548	629

**Table 4 polymers-15-02340-t004:** Concrete mixes parameters.

Mix	Hybridization Ratio (SF–PPF %)	Compressive Strength (Mpa)	Splitting Strength (Mpa)	Flexural Strength (Mpa)	Modulus of Elasticity (Mpa)
M1	0–0%	37.3	2.75	3.55	31,378.2
M2	100–0%	38.2	3.9	7	32,897.4
M3	75–25%	36.6	3.75	6.7	30,251.6
M4	50–50%	33.4	3.5	6.4	28,873.4
M5	25–75%	31.5	3.3	4.3	26,872.1
M6	0–100%	28.4	2.9	4	23,428.7

**Table 5 polymers-15-02340-t005:** CDP input data for concrete.

Concrete Type	Dilation Angle	Eccentricity	$f_{b0}/f_{c0}$	K	Viscosity Parameter
Normal	35	0.1	1.16	0.667	0.0079
Hybrid	37	0.1	1.16	0.667	0.005

**Table 6 polymers-15-02340-t006:** Damage representation for all numerical all six models.

Case	Steel Stress Intensity (σ/σy) 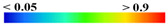	Tension Damage Coefficient (dt) 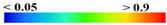
M1B	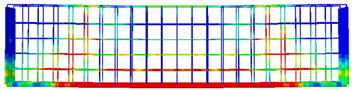	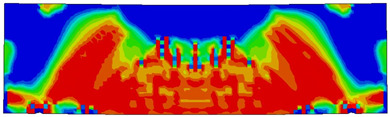
M2B	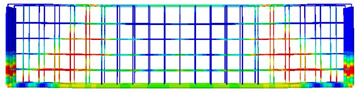	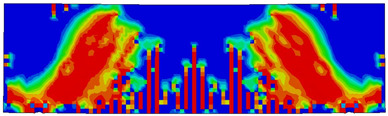
M3B	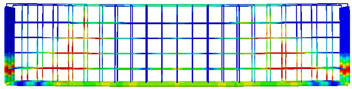	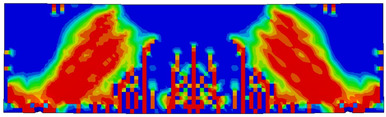
M4B	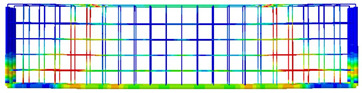	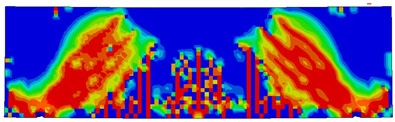
M5B	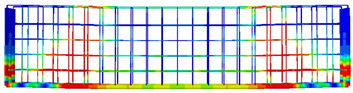	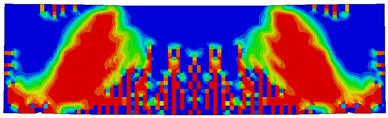
M6B	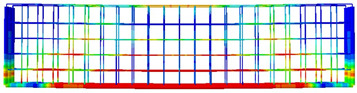	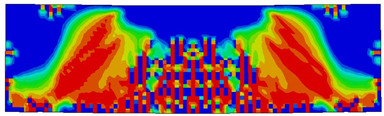

## Data Availability

The datasets which were generated during and analysed during the current study are available in the main manuscript, any additional details can be obtained from the authors.

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
