# Peer review of "Experimental and Numerical Analysis of Steel-Polypropylene Hybrid Fibre Reinforced Concrete Deep Beams"

_polymers, 2023, doi:10.3390/polym15102340_

Round 1

Reviewer 1 Report

The authors utilized different fiber in reinforced concrete deep beams. Two beams were tested and rest was carried out in abaqus. The paper is generally good but it needs improvement. Followings should be carried out before acceptance:

The abstract should contain important results of the study.

The sentence givien in line 47 "The shear-span-to-depth ratio is found to have a 46

greater impact on the shear capacity than the span-to-depth ratio in parametric research" can be strengthen using following reference: "numerical evaluation of effects of shear span, stirrup spacing and angle of stirrup on reinforced concrete beam behaviour"

The novelty statement is not clear in introduction. The authors should declare clearly what has been done different from current study. What this study offer for novelty.

Tensile properties of concrete with fiber is different from plain concrete. The model utilized by authors are not valid for concrete with fibers. The authors should discuss this.

The reviewer thinks that compression and tension models are wrong. Please check the recommendation of RILEM:test and design methods for steel fibre reinforced concrete

The sentence given in line 73 "Steel fibres improve the concrete's strength and ductility, while polypropylene fibres make it more durable and crack-resistant." can be strengthen with followings: improvementg performance of reinforced concrete beams produced with waste lathe scraps; performance assessment of fiber-reinforced concrete produced with waste lathe fibers; performance evaluation of fiber-reinforced concretes produced with steel fibers extracted from waste tire; investigation on improvement in shear performance of reinforced-concrete beams produced with recycled steel wires from waste tires; geopolymer concrete with high strength, workability and setting time using recycled steel wires and basalt powder

The reason for selecting design mixture should be added.

Compare your results with existing studies

The use of CDP model in concrete can be strengthen following studies:experimental and numerical investigations of steel fiber reinforced concrete dapped-end purlins; Experimental, analytical and numerical investigation of pultruded GFRP composite beams infilled with hybrid FRP reinforced concrete;

How this recycled materials for this study is obtained?

Add sieve analysis results in Figure.

What is chemical properties of cement

Novelty is not clear. Very same studies are already exists. What is the difference?

Add photos for test setup?

Add some summary for conclucision

Add recent studies on this subject to introduction. There are many studies on the introduction for this topic.

Conclusion should be improved. The recommendation consdiering all test should be given for engineers.

To sum up, the main problem is modeling of concrete with fiber. The authors modeled concrete with fiber smilar to plain concrete. Unfortunutaly this is completely wrong.

Reviewer 2 Report

Reviewer Comment:

The paper titled Numerical plastic analysis of hybrid polymers reinforced concrete deep beams.

The authors of this study investigated the effects of variable mixtures based on steel fibers as well as polypropylene fibers on the behavior of reinforced concrete beams, both experimentally and numerically.  Thus, two types of beams were tested with the same dimensions, the first one made from a standard concrete mix (without fibers) while the second one was made from a hybrid polymer concrete. The authors also used the ABAQUS software for numerical simulation. The article is well organized and the work is very interesting. Therefore, I suggest that the manuscript must be accepted after major revision.

General Comments:

1.      The manuscript is clear, relevant for the field, and presented in a well-structured manner.

2.      The manuscript is scientifically sound, the experiments are appropriately designed, and the methods are well described.

Some suggestions are as follows:

1.       In my opinion the title can be changed;

2.       The Abstract does not contain any results values. The abstract section must be revised to give a brief explanation of the importance, investigations and outcomes with the advantages/significance of this research study. Also, the novelty of the study should be reflected in the abstract.

3.       Please add author references in the introduction section and it is necessary to mention clearly the originality and novelty of this study in the end of this section.

4.       Several figures are not clear, for example: Fig1, Fig 13.

5.       Please insert the units in figure 9.

6.       Please insert more details in figure 10 and indicate more precisely what each figure means, for example a, b, c and d.

7.       Please, it is necessary that the authors present and more discussed the results and compare with other work in the literature.

8.       The authors are not showing any other data as a comparison or reference value, to be compared with their generated data.

9.       A complete revision of the document is necessary. Improved bibliography, there are more recent references not cited.

Minor editing of English language required.

Reviewer 3 Report

Reviewer comments 

Manuscript ID: Polymers-2371886

Title: Numerical plastic analysis of hybrid polymers reinforced concrete deep beams

Journal: Polymers

The Presented study focuses on investigating the behavior of simply supported reinforced concrete deep beams by using different mixes of steel and polypropylene fibers. The study uses both experimental and numerical analysis to examine the effects of the fiber mixes on the beams.

The paper lies within the scope of the journal, however, I think it needs some improvement. I would suggest the following revision on the paper.

1-      The novelty of this article should be emphasized and should be addressed clearly in the introduction and regarding what you have developed, found, or improved compared with the other similar research’s published before.

2-      In the abstract lines (9, 10) reformulate sentence.

3-      Line 154: correct this sentence “This study aims of this study are to investigate…. deep beams, both experimentally and numerically”

4-      Line 158: correct this sentence “The novelty of the study lies in the investigation …” and avoid long sentence.

5-      Avoid repeating sentences in the manuscript, in particular in the abstract and in the introduction (ex ... lines 9-11, 154-156).

6-      Typos and grammar problems need to be corrected properly, authors should carefully check through the manuscript before submitting a revision

7-      Most of the figures are of bad quality (Figs. 1, 5, 10, 13 …), there are pixels around lines, leaving an impression of a “dirty” figures. Most of the figures need to be improved and expand the figures legends and axes to make them more visible and readable.

8-      Figure 2: some details missed

9-      In figure 9-a, the authors present the compression tests of samples with 1400x400x150 dimension, how about buckling during compression tests. Authors should clarify this point.

10-  In figure 9-b, the authors present the tension tests with the same samples, how the samples are fixed on the tensile machine. Authors should clarify this point.

11-  Authors should clarify how parameters in table 5 are determined

12-  How the authors adjusted mesh size to the geometry of presented model? What mesh refinement algorithm was used in order to obtain optimal mesh size for presented analyses? If mesh refinement was performed manually, appropriate Figure presenting i.e. stress-mesh size dependency should be included in the manuscript.

The reviewer recommends that the author do major revision to the. Also, the reviewer would recommend that the authors proofread the article thoroughly for typos and grammatical errors.

The reviewer recommends that the author do major revision to the. Also, the reviewer would recommend that the authors proofread the article thoroughly for typos and grammatical errors.

Round 2

Reviewer 1 Report

The authors ignored all most of all questions of the reviewer.

In this current form. the paper can not be accepted. 

The authors shoud revise their studies based on the suggestion of reviewer.

Moreover, there is no single imrovement. There is no highlted text.

For example, the reviewer asked to add important results to abstract. But the authors add as response for old whole abstract.

The reviewer asked to improve novelty. The authors responsded with old whole introduction.

They ignored all suggested studies

They can not use standard astm for fiber tensile. If they insisted on this, the authors should provide 2-3 references from high quality journals in which the authors utilized fibers with normal concrete behavior in their numerical analyses

Reviewer 2 Report

Accept in present form

Author Response

The authors express their sincere thanks and appreciation to the editor and reviewer for taking the time to review this manuscript.

Reviewer 3 Report

Most of the points have been correctly addressed by the authors.

ok

Author Response

(The authors gave the same response as above.)
